# SuperVision: Self-Supervised Super-Resolution for Appearance-Based Gaze Estimation

**Galen O'Shea**                                          GalenOShea@cmail.carleton.ca

**Majid Komeili**                                         Majid.Komeili@carleton.ca

*Carleton University, Canada*

## Abstract

Gaze estimation is a valuable tool with a broad range of applications in various fields, including medicine, psychology, virtual reality, marketing, and safety. Therefore, it is essential to have gaze estimation software that is cost-efficient and high-performing. Accurately predicting gaze remains a difficult task, particularly in real-world situations where images are affected by motion blur, video compression, and noise. Super-resolution (SR) has been shown to remove these degradations and improve image quality from a visual perspective. This work examines the usefulness of super-resolution for improving appearance-based gaze estimation and demonstrates that not all SR models preserve the gaze direction. We propose a two-step framework for gaze estimation based on the SwinIR super-resolution model. The proposed method consistently outperforms the state-of-the-art, particularly in scenarios involving low-resolution or degraded images. Furthermore, we examine the use of super-resolution through the lens of self-supervised learning for gaze estimation and propose a novel architecture "SuperVision" by fusing an SR backbone network to a ResNet18. While only using 20% of the data, the proposed SuperVision architecture outperforms the state-of-the-art GazeTR method by 15.5%.

**Keywords:** Gaze Estimation, Super-Resolution, Self-Supervision, GANs

## 1. Introduction

Gaze estimation has a wide variety of important applications in research and in the real world. It serves as a valuable tool in fields such as cognitive science and psychology, facilitating investigation into cognitive impairments including dementia (Mengoudi et al., 2020), autism (Anzalone et al., 2019), and attention-deficit/hyperactivity disorder (Pishyareh et al., 2015). In the realm of healthcare, gaze estimation proves beneficial for identifying visual impairments such as age-related macular degeneration (Yow et al., 2017). Additionally, it holds the potential to significantly improve the quality of life for people living with paralysis, amyotrophic lateral sclerosis, and other locked-in disabilities as it allows them to interact through eye movements (Liu et al., 2010). In the automotive industry, gaze estimation could improve car safety. With this technology, cars could monitor driver attention to detect signs of fatigue, or determine potential dangers on the road ahead (Ishikawa, 2004). Furthermore, gaze estimation can be applied in virtual reality (Padmanaban et al., 2017) as well as analyzing consumer attention in marketing research (Modi and Singh, 2022). Given the breadth of important applications, it is evident that there is a need for high-quality gaze estimation technologies.

Gaze estimation research is often divided into two primary methodologies: model-based and appearance-based approaches. While model-based methods require specialized hardware, appearance-based methods only require a camera (Hansen and Ji, 2009). Appearance-based methods compensate for the absence of specialized hardware by using large and complex machine learning algorithms to extract information (Zhang et al., 2015; Ali and Kim, 2020; Bao et al., 2021). Formally, appearance-based gaze estimation can be delineated as a supervised learning task, wherein a machine learning model is trained on a dataset of faces labeled by known gaze directions. The objective is to learn a mapping between the input image containing facial features and output gaze direction. The gaze direction can be represented in various ways, but a prevalent approach involves using a two-dimensional gaze vector to denote the gaze's pitch and yaw angles. Although there have been improvements in appearance-based gaze estimation, accurately predicting gaze remains a difficult task, particularly when done outside of controlled environments and without curated datasets (Alberto Funes Mora and Odobez, 2014; Xiong et al., 2014; Sun et al., 2015; Funes Mora et al., 2014).

One of the fundamental steps in any data science workflow involves data preprocessing. Real-world data is often characterized by inconsistencies and noise, requiring a transformation to enhance the learning process for machine learning (ML) models. In the context of tabular data, preprocessing might involve removing null values, encoding categorical data, and performing feature scaling. Similarly, natural language processing (NLP) uses preprocessing techniques such as lower-casing, tokenization, and punctuation mark or stop word removal. In computer vision (CV), we might resize, normalize or even apply data augmentation through flipping, rotating, warping, and modifying colour intensity. While these CV methods might help the model generalize, none attempt to denoise or remove irrelevant pixels in a fashion similar to removing null values from tabular data or removing punctuation marks and stop words in NLP.

Super-resolution (SR) is a technique employed to enhance the resolution of an image beyond its original quality, and it has been shown to enhance image quality from a visual perspective (Ledig et al., 2017; Wang et al., 2018, 2021b,a; Liang et al., 2021; Saharia et al., 2021). SR has been used in various applications, including medical imaging (Shi et al., 2013), remote sensing (Ling and Foody, 2019), surveillance, and video processing (Liu et al., 2022). From a qualitative standpoint, SR has demonstrated the ability to create visually appealing facial images, however, its usefulness for gaze estimation has yet to be investigated.

Despite the advancements in gaze estimation techniques, there is still room for improvement. It has been suggested that deep learning gaze estimation models could benefit from datasets that contain more pixels within each eye patch (Ali and Kim, 2020). While this would help to improve gaze estimation models, it requires the creation of an entirely new dataset, which can be time-consuming and impractical. Moreover, current approaches in gaze estimation lack the preprocessing steps necessary to improve the efficacy of ML models. To address these issues, we propose a two-step framework using SR to preprocess images, thereby, enhancing the quality of existing datasets with the goal of using high-resolution images to improve current gaze estimation approaches.

Furthermore, we examine SR through the lens of self-supervised learning for gaze estimation. Self-supervised learning allows a machine learning model to learn from unlabelled

data, which can be easier and less expensive to acquire than labeled data. For example, a self-supervised model might be trained to predict the rotation or colour of an image, which can be done without the need for manual annotation or labelling. Once the self-supervised model is trained, it can be fine-tuned for a specific task using a smaller amount of labelled data. Gaze estimation, like many traditional supervised learning tasks, requires a large amount of labelled data. A sample-efficient model opens up the possibility of building gaze models for scenarios where obtaining a large labelled gaze dataset is challenging, such as gaze estimation in infants (Franchak et al., 2016), older adults (Chapman and Hollands, 2006) or animals (Wiltschko et al., 2015; Ogura et al., 2020; Guo et al., 2003). Therefore, we propose using self-supervised learning for an SR-Gaze model "SuperVision" to improve end-to-end appearance-based gaze estimation.

The contributions of this paper are as follows:

1. We rigorously examine the effect of SR, when used as a preprocessing step for appearance-based gaze estimation, and show that SR models may not preserve the gaze direction.

2. We propose a two-step framework based on SR and achieve state-of-the-art results on the MPIIFaceGaze (Zhang et al., 2015) dataset. Additionally, the performance of the proposed method is evaluated on multiple lower-resolutions and various degradations, consistently outperforming previous methods.

3. Furthermore, we propose a novel architecture named "SuperVision" by fusing an SR backbone network to a ResNet18 (with skip connections). The proposed SuperVision method outperforms the state-of-the-art method GazeTR (Cheng et al., 2021) by 15.5%, while only using 20% of the dataset. The proposed SuperVision method opens up the possibility of building gaze models for scenarios where obtaining a large labelled gaze dataset is challenging, such as with infants, older adults, and various types of animals.

## 2. Related Work

### 2.1. General Approaches

In the field of gaze estimation research, there are two primary approaches: model-based and appearance-based methods (Hansen and Ji, 2009). Model-based methods aim to predict gaze by using either a geometric three-dimensional eye model that analyzes the corneal reflection (Nakazawa and Nitschke, 2012), or shape-based methods that utilize the pupil centre (Valenti et al., 2011) or contours of the iris (Funes Mora et al., 2014). Advancements in corneal-reflection techniques have enabled the transition from fixed head positions (Morimoto et al., 2002) to multiple head poses and lighting conditions (Zhu et al., 2006). However, due to the complexity of these methods, researchers often employ costly and specialized hardware (e.g., depth sensors) that may not be practical for general-purpose gaze estimation (Alberto Funes Mora and Odobez, 2014; Xiong et al., 2014; Sun et al., 2015; Funes Mora et al., 2014). Additionally, although these methods exhibit outstanding performance in a controlled laboratory setting, they are less reliable in low light conditions and unconstrained environments.

Appearance-based methods frame gaze estimation as a regression problem by mapping gaze images to a corresponding gaze vector, while avoiding the need for specialized hardware and using only a camera. However, early appearance-based methods required time-consuming head-pose calibration for each participant, which led to research on reducing the number of training examples using semi-supervised Gaussian regression methods (Williams et al., 2006) and finding an optimal set of training samples using adaptive linear regression (Lu et al., 2014b). Despite these efforts, calibration remained insufficient as models failed to generalize to new subjects and head poses, leading to research on solving head pose and subject-related issues using a pose-based clustering method (Sugano et al., 2008) and compensating for bias via regression (Lu et al., 2014a). To address free head motion, eye image synthesis was later employed (Lu et al., 2015). Generalization problems were further handled using cross-subject training methods (Mora and Odobez, 2013) and learning-by-synthesis methods (Sugano et al., 2014).

## 2.2. Deep Learning Approaches

Early approaches to gaze estimation faced significant challenges in adapting to new subjects and positions despite efforts to enhance their performance. As a result, research shifted to deep learning approaches such as the Multimodal Convolutional Neural Network (CNN), which concatenate eye images with a head pose estimate (Zhang et al., 2015). In subsequent work, the authors designed a Visual Geometry Group (VGG) inspired architecture that extended their previous work (Zhang et al., 2017a). Other methods included dilated-convolutions to extract high-level features without reducing spatial resolution (Chen and Shi, 2018). In related studies on multi-stream CNNs, researchers adopted data fusion, which involved merging datasets while maintaining separate validation sets to evaluate the performance of each dataset independently (Ali and Kim, 2020). Although state-of-the-art performance was achieved, the variation in accuracy across datasets was attributed to differences in the resolution of the eye patch. The researchers suggested that deep learning models for gaze estimation could benefit from higher accuracy if the datasets had more pixels within each eye patch (Ali and Kim, 2020). Studies have also indicated that combining eye feature maps can enhance the accuracy of gaze estimation. For instance, a study utilized a VGG-16 network per eye, with concatenated downstream features (Fischer et al., 2018a). Other works employed a quad-stream architecture to extract singular and joint features from both eyes (Chen and Shi, 2018). Subsequently, researchers explored the use of the attention mechanism to extract joint eye features. In one novel approach (Bao et al., 2021), researchers proposed an Adaptive Feature Fusion technique that stacked eye feature maps based on their similarity, using a self-attention mechanism. More recently, transformer architectures have emerged as dominant in the field, including for gaze estimation. The current state-of-the-art model, GazeTR, uses a hybrid CNN and transformer architecture (Cheng and Lu, 2021).

In the past, the predominant approach to solving the gaze estimation problem has been to develop complex models, with limited attention given to improving the quality of the data. While a few studies have utilized generative adversarial networks (GANs) to enhance data quality by improving lighting conditions (Kim and Jeong, 2020) or removing artifacts from glasses (Rangesh et al., 2020), these niche approaches have demonstrated

limited practical success, as they do not address the fundamental challenge of obtaining high-quality datasets.

## 2.3. Super-Resolution

The first application of GANs for super-resolution was in SRGAN, which outperformed prior methods and achieved state-of-the-art results (Ledig et al., 2017). The authors attributed this, in part, to their use of a perceptual loss function that accounted for perceptual similarity instead of just similarity in pixel space. At that time, a common problem with super-resolution was the presence of artifacts when upsampling. ESRGAN addressed this issue by identifying that batch normalization layers tended to create unwanted artifacts (Wang et al., 2018). They also improved the perceptual loss function and used Residual-in-Residual Dense Blocks to generate more realistic images consistently. Additionally, they later proposed REAL-ESRGAN, which incorporated a u-net discriminator and spectral normalization (Wang et al., 2021b). They also developed a complex degradation method that used only synthetic data for real-world degradations.

Face restoration is often challenging because it requires prior knowledge, such as facial geometry, to restore realistic details. GFP-GAN utilized generative facial priors in the image restoration process and achieved realistic details and state-of-the-art results (Wang et al., 2021a). Other attempts at blind image restoration have employed transformers. For example, one study utilized a shifted window transformer as a deep feature extractor in a model called SwinIR, which achieved state-of-the-art results with up to 67% fewer parameters (Liang et al., 2021). However, GANs for super-resolution are becoming less popular, as the current state-of-the-art is based on an iterative refinement method (Saharia et al., 2021), which significantly outperforms previous works.

While most existing methods typically employ classical degradation methods such as downsampling to generate low-resolution images, Real-ESRGAN used synthetic data and a complex degradation model that aimed to simulate real-world complex degradations (Wang et al., 2021b). Their degradation model combined multiple classical degradations, including Gaussian filters for blurring, downsampling using interpolating methods, adding Gaussian, colour, and other types of noise, and reducing quality through JPEG compression. However, other researchers found that the degradation model used by Real-ESRGAN lacked diversity, and they addressed this issue by expanding the model through random shuffling of the process, adding different levels of noise and compression, and introducing processed camera sensor noise and RAW image noise (Zhang et al., 2021).

## 2.4. Self-Supervised Learning

Self-supervised learning has emerged as a powerful technique for learning rich and meaningful representations from unlabeled data. This technique involves training a model to predict a pretext task from the input data, which then results in learning useful representations that can be transferred to downstream tasks. One popular form of self-supervised learning is contrastive learning, which learns representations by contrasting positive and negative pairs of samples (He et al., 2019; Grill et al., 2020; Tian et al., 2020). SimCLR (Chen et al., 2020) has been one of the most successful and extensively used approaches in this field. It has achieved state-of-the-art results on various benchmarks, including ImageNet and COCO.

The researchers have demonstrated that this technique is effective in pretraining models for several downstream tasks, such as object detection, instance segmentation, and semantic segmentation.

## 3. On the Usefulness of SR for Gaze Estimation

### 3.1. Not All SR Methods Preserve Gaze

Prior studies have suggested that using face images with higher pixel density in the eye regions could enhance the accuracy of gaze estimation using deep learning (Ali and Kim, 2020). However, instead of curating a new high-resolution dataset, we propose using super-resolution (SR) to enhance the quality of existing datasets. Super-resolution refers to the process of increasing the resolution of an image by recovering or generating high-resolution images from low-resolution inputs. It has been shown to be useful in various computer vision tasks, but its usefulness for gaze estimation has not been studied. In particular, while SR can increase detail and clarity, it is not apparent if the SR process alters the gaze in the resulting image. Along this line, we examine super-resolution's effect on gaze estimation using two different SR methods.

Although several SR models are available, we focused on GFP-GAN (Wang et al., 2021a) and SwinIR (Liang et al., 2021) due to their high performance in Peak-Signal-To-Noise-Ratio (PSNR) and Structural-Similarity-Index-Measure (SSIM), which are predominant in the evaluation of generative image techniques. These models adopt distinct approaches to SR. GFP-GAN is specifically designed for facial image restoration. The method is trained on the high-resolution dataset FFHQ (Karras et al., 2019) and leverages facial priors acquired from the dataset for image restoration. In contrast, SwinIR takes a general approach as it uses a transformer to focus on blind-SR. Blind-SR attempts to enhance the resolution of a low-resolution image without prior knowledge of the degradation model or the high-resolution image. SwinIR is therefore trained on two general datasets, DIV2K (Ignatov et al., 2019) and Flickr2K (Timofte et al., 2017), which are not specifically facial datasets. These unique approaches may offer valuable insights about the impact of SR on gaze estimation.

When choosing a model for gaze estimation, we opted for the Full-Face model (Zhang et al., 2017b), which is widely acknowledged as the baseline for research in appearance-based gaze estimation. Furthermore, Full-Face was used as the original baseline on the MPIIFaceGaze dataset (Zhang et al., 2015), which is a popular benchmark for gaze estimation. MPIIFaceGaze is comprised of 45,000 images of size $448 \times 448$, gathered from 15 subjects of different ethnicities under natural lighting conditions. The dataset was preprocessed using the same methodology as GazeTR (Cheng and Lu, 2021), thereby, cropping facial images by their bounding box. Additionally, to align with many previous works, we opted to use a two-dimensional gaze vector representing pitch and yaw as the ground-truth label for each image. This vector represents the pitch and yaw of the gaze transposed from the relative three-dimensional gaze collected in MPIIFaceGaze. Given the two-dimensional ground-truth, angular error can be used as an insightful evaluation metric by measuring the difference between prediction and ground-truth in degrees. For reproducibility and a fair comparison to the gaze models, we used GazeTR's hyperparameters (Zhang et al., 2015). The batch size was 512, the number of epochs was left at 100, the learning rate was 0.0005

with a warm up of 5 epochs, a decay of 0.5 and decay step of 60, the optimizer used was Adam with betas 0.9, 0.95 and the criterion used was Absolute Error Loss (L1 Loss).

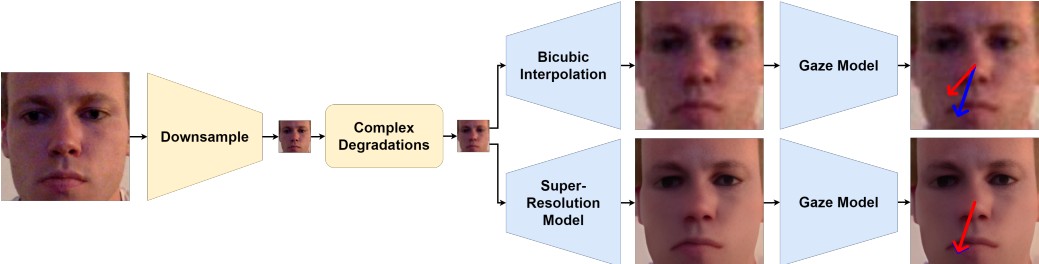

Figure 1: Starting with a high-resolution image, the initial step involves downsampling to a lower resolution and optionally applying intricate degradations. The two-step framework comprises the first stage, where the image undergoes preprocessing through super-resolution (or bicubic interpolation to establish a baseline), followed by the second stage, which involves performing gaze estimation on the preprocessed image.

To evaluate the efficacy of the proposed method, the images were first downsampled from size $448 \times 448$ to size $112 \times 112$ to simulate low-resolution images. We also use BSR-GAN's image degradation method (Zhang et al., 2021), which applies a multitude of degradations (e.g., JPEG compression, Gaussian blur, noise, etc.) producing complex degradations to simulate real-world degradations. Subsequently, we used the proposed two-step framework which first enhances the degraded images using super-resolution and then uses those enhanced images for gaze estimation (Figure 1). In the first step, we applied one of the pretrained SR models, GFP-GAN (Wang et al., 2021a) or SwinIR (Liang et al., 2021), to upscale the images back to size $448 \times 448$. To ensure that upscaling did not serve as a confounding variable, we also upscaled the dataset using bicubic interpolation to create a baseline. In step two, we trained and evaluated Full-Face with each of the three new datasets using a leave-one-out cross-validation approach. This involved training on 14 of the 15 participants for 100 epochs and testing on the excluded participant. We repeated this process for each participant to calculate the average angular error and present the model with the highest test performance for each dataset. The aim of this experiment was to assess how various SR methods compare to an interpolation baseline when applied to low-resolution images of size $112 \times 112$. Furthermore, we evaluate with and without complex degradations and report the results in Table 1. The training time was approximately 3 hours per participant on a Tesla V100.

Table 1 shows the results of experiments with and without complex degradations. Without degrations, SwinIR-4x exhibited a 2.1% improvement over the baseline, in contrast GFP-GAN performed 9.3% worse. In the degraded scenario, all interpolation methods achieved mediocre results in relation to the non-degraded scenario, which can be attributed to the noise caused by complex degradations. Once again, SwinIR-4x outperformed the baseline by a greater margin of 6.8% while GFP-GAN under-performed relative to the baseline. The results indicate that while GFP-GAN can generate visually sharp images, it is inadequate

| Degradation Type | Interpolation Type | Angular Error [°] | Change [%] |
|---|---|---|---|
| | Interpolation | 4.20 | - |
| None | GFP-GAN-4x | 4.59 | -9.29% |
| | SwinIR-4x | **4.11** | **2.14** |
| | Interpolation | 5.47 | - |
| Complex Degradations | GFP-GAN-4x | 5.76 | -5.30% |
| | SwinIR-4x | **5.10** | **6.76** |

Table 1: Comparison of different SR methods for gaze estimation using Full-Face. GFP-GAN performs worse than a simple interpolation while SwinIR improves gaze estimation and has increased relative performance over the baseline on degraded data.

for the task of gaze estimation. On the other hand, SwinIR outperformed the baseline with both regular and degraded images. The most interesting finding is that SwinIR achieved a 5.3% improvement in relative performance indicating that it was more effective in degraded scenarios, emphasizing the applicability of SR in real-world situations. While this experiment shows a large contrast in SR methods, it has also demonstrated that SR can be an effective tool for gaze estimation.

## 3.2. Looks Can Be Deceiving

As demonstrated in Table 1, not all SR methods are suitable for task dependent image restoration. GFP-GAN's poor gaze estimation performance might be attributed to the use of facial priors, which bias the model into hallucinating facial features from noise. Additionally, "in the face of uncertainty", GFP-GAN's dependence on facial priors resulted in a multitude of images having a centralized gaze. This hallucination of gaze is most evident when the input images are significantly degraded (see Figure 2), indicating that GFP-GAN is susceptible to mode collapse. Unlike other restoration approaches, GFP-GAN was trained on the FFHQ dataset (Karras et al., 2019), which predominately contains people looking directly at the camera. Since GFP-GAN does not attempt to maintain the original gaze and is influenced by facial priors, the reconstructed faces have a centralized gaze and thus "looks can be deceiving".

## 3.3. Enhancing Gaze Estimation with SR

In our second experiment, to further enhance gaze estimation performance, Full-Face is replaced with a state-of-the-art model, GazeTR (Cheng and Lu, 2021). GazeTR was proposed as two architectures, one is based on a pure transformer architecture while the other hybridizes the transformer with a CNN. We will be focusing on the hybrid architecture since it demonstrated considerably better performance (Table 2). Considering the poor performance of GFP-GAN, only SwinIR will be used for the remaining experiments. We demonstrate that the proposed two-step framework surpasses GazeTR and produces state-of-the-art results on the MPIIFaceGaze dataset (Zhang et al., 2015). To reproduce claims by the authors, GazeTR was pretrained on ETH-XGaze, which is a dataset consisting of

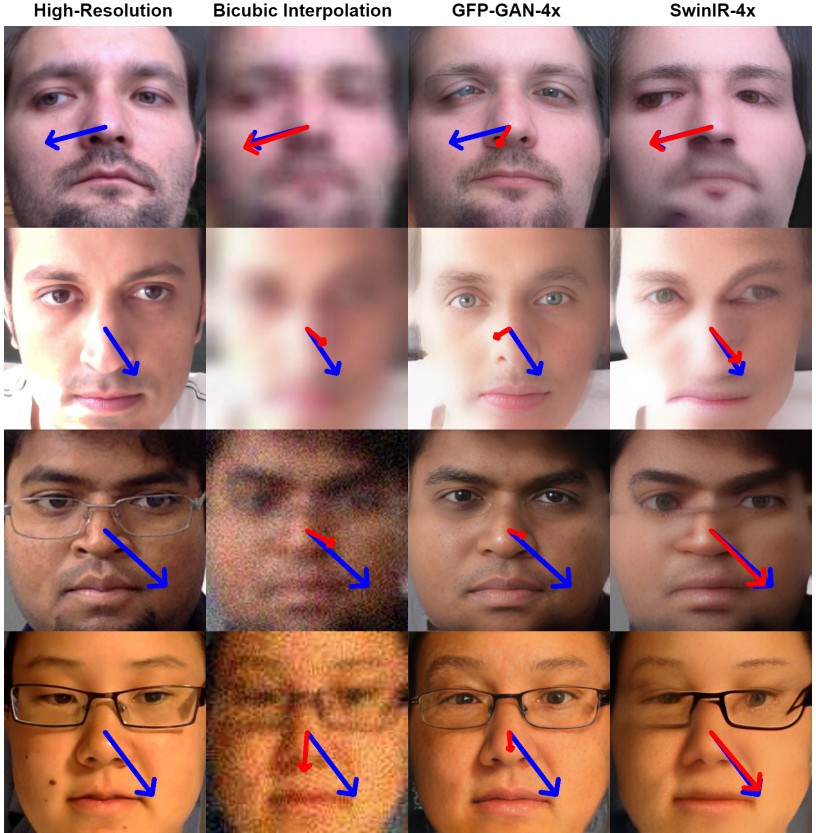

Figure 2: The left most column depicts original images which are then degraded and down-sampled to size $112 \times 112$ in the second column. The third and fourth columns depict the degraded images restored to $448 \times 448$ using GFP-GAN-4x and SwinIR-4x, respectively. The blue arrow shows the ground truth gaze vector while red blue arrow shows the predicted gaze.

over one million high-resolution gaze images presented in multiple head poses. The dataset was collected from 110 subjects using multiple SLR cameras, lighting conditions, and a calibrated ground truth (Zhang et al., 2020).

Since the GazeTR was originally evaluated on images of size $224 \times 224$, the proposed two-step framework was first evaluated by upsampling from $224 \times 224$ to $448 \times 448$ using SwinIR-2x and then downsampling back to $224 \times 224$ before being passed to GazeTR. By downsampling after SR, we ensure that model complexity is not a confounding variable when comparing to the state-of-the-art. Furthermore, it is theorized that downsampled SR would inherently lose image quality, so we evaluate the two-step framework again but at size $448 \times 448$ by not downsampling and compare it to a baseline of bicubic interpolation. Finally, GazeTR is used on the original $448 \times 448$ high-resolution data providing an upper limit on performance. The training time was approximately 6 hours at size $224 \times 224$ and 12 hours at size $448 \times 448$ per participant on a Tesla V100.

| Method | Interpolation Input Size | Gaze Input Size | Angular Error [°] |
|---|---|---|---|
| Full-Face (Zhang et al., 2017b) | - | 224 | 4.93 |
| RT-Gene (Fischer et al., 2018b) | - | 224 | 4.66 |
| Dilated-Net (Chen and Shi, 2019) | - | 224 | 4.42 |
| Gaze360 (Kellnhofer et al., 2019) | - | 224 | 4.06 |
| CA-Net (Cheng et al., 2020) | - | 224 | 4.27 |
| Mnist (Zhang et al., 2015) | - | 224 | 6.39 |
| GazeNet (Zhang et al., 2017a) | - | 224 | 5.76 |
| GazeTR-Pure (Cheng and Lu, 2021) | - | 224 | 4.74 |
| GazeTR-Hybrid (Cheng and Lu, 2021) | - | 224 | 4.00 |
| (Ours) SwinIR-2x Downsampled | 224 | 224 | **3.94** |
| Bicubic Interpolation | 224 | 448 | 3.99 |
| (Ours) SwinIR-2x | 224 | 448 | **3.90** |
| Original High-Resolution | 448 | 448 | 3.94 |

Table 2: Comparison with prior works. The interpolation input size column denotes the size of an image entering the super-resolution step of the framework, while the gaze input size column denotes the size of the input into the gaze model, for instance, 224 denotes $224 \times 224$.

The first section in Table 2 demonstrates the performance of prior works on images of size $224 \times 224$, while the second section demonstrates results from our second experiment compared to the current state-of-the-art GazeTR (Hybrid). The results show "SwinIR-2x Downsampled" achieved an angular error of $3.94°$ and a 1.5% improvement in performance compared to GazeTR's angular error of $4.00°$. This result shows that SR can still provide additional performance even to state-of-the-art models such as GazeTR. The bicubic interpolation baseline achieved an angular error of $3.99°$ indicating that increasing the model complexity and image size to $448 \times 448$ does not play a significant role in performance since GazeTR achieved an angular error of $4.00°$ at size $224 \times 224$. Interestingly, SwinIR-2x achieved an angular error of $3.90°$ and demonstrated a 2.3% improvement over the bicubic interpolation baseline, showing greater relative performance when the super-resolved image is not compressed back down to size $224 \times 224$. This contrast in performance to bicubic interpolation also suggests that the benefits of SR go beyond just making an image larger and can likely be attributed to denoising properties as SR increases the resolution. Furthermore, SwinIR-2x achieved a 1% improvement in performance over the original $448 \times 448$ high-resolution data (angular error of $3.94°$). This surpasses the theorized performance upper limit even when starting with lower-resolution data.

### 3.4. Low-Resolution and Degraded Images

To further elaborate on the findings from Table 2, SR was examined with two additional experiments. The first experiment expanded on the idea that SR can improve gaze estimation when using lower resolutions. By examining SR compared to a baseline on lower

resolutions we can solidify the notion that SR preprocessing is more than just the increase of pixels but rather the denoising of low quality images. The second experiment looked at the denoising of SR in a different light by evaluating SR on degraded data. This experiment analyzed SR beyond simply lower-resolution images by introducing complex degradations. Additionally, while reproducing the results of GazeTR (Cheng and Lu, 2021), it was discovered that a significant portion of its performance was attributed to its pretraining on the ETH-XGaze dataset (Zhang et al., 2020). Thus, the effect of pretraining the gaze model was also investigated.

To simulate a low-resolution setting, images were downsamped to $112 \times 112$ and also $56 \times 56$ (4x and 8x smaller than the original dataset, respectively) to evaluate gaze estimation performance in a more extreme setting. SR preprocessing is evaluated against a bicubic interpolation baseline and the results of these low-resolution experiments are shown in Table 3. When evaluating SR preprocessing on complex degradations, BSR-GAN's complex degradation method (Zhang et al., 2021) was used, which aims to simulate real-world degradations such as motion blur, pixelation, and compression. Furthermore, these experiments were repeated for GazeTR with and without pretraining (see Table 3 and Table 4). Similar training times were found as the previous experiments with the inclusion of training on images of size $56 \times 56$, which took approximately 1.5 hours. While it is not surprising that pretrained models outperformed non-pretrained, SR exhibited improved performance relative to the interpolation baseline, irrespective of pretraining, initial image resolution or the addition of complex degradations. The experiments carried out, emphasize the application of the SR two-step framework for gaze estimation, particularly in scenarios where facial images are low-resolution or have been affected by real-world degradations such as motion blur, pixelation, and compression.

| Pretraining | Interpolation Type | Interpolation Input Size | Gaze Input Size | Angular Error [°] | Change [%] |
|---|---|---|---|---|---|
| No | Interpolation | 56 | 224 | 4.81 | - |
| | SwinIR-4x | 56 | 224 | **4.76** | **1.04** |
| | Interpolation | 112 | 448 | 4.53 | - |
| | SwinIR-4x | 112 | 448 | **4.48** | **1.10** |
| Yes | Interpolation | 56 | 224 | 4.31 | - |
| | SwinIR-4x | 56 | 224 | **4.22** | **2.09** |
| | Interpolation | 112 | 448 | 4.24 | - |
| | SwinIR-4x | 112 | 448 | **4.21** | **0.71** |

Table 3: Using GazeTR as the gaze model and lower-resolution images. The interpolation input size column denotes the size of an image entering the super-resolution step of the framework, while the gaze input size column denotes the size of the input into the gaze model.

| Pretraining | Interpolation Type | Angular Error [°] | Change [%] |
|:---:|:---:|:---:|:---:|
| No | Interpolation | 5.40 | - |
| | SwinIR-4x | **5.33** | **1.30** |
| Yes | Interpolation | 5.37 | - |
| | SwinIR-4x | **5.20** | **3.17** |

Table 4: Using GazeTR as the gaze model and degraded 112×112 images. The interpolation input size column denotes the size of an image entering the super-resolution step of the framework, while the gaze input size column denotes the size of the input into the gaze model.

## 4. A Self-Supervised Approach Based on Super-Resolution

As in many computer vision tasks, it is often unfeasible to collect large gaze datasets due to challenges in data acquisition. In some cases it might be more challenging when dealing with small sample groups such as collecting gaze datasets for infants (Franchak et al., 2016) or older adults (Chapman and Hollands, 2006). Furthermore, it can also be quite challenging working with uncooperative groups such as gaze estimation in animals (Wiltschko et al., 2015; Ogura et al., 2020; Guo et al., 2003). Research has previously shown that self-supervised learning can be an effective method for pretraining a backbone network on unlabelled data for the purpose of domain adaptation. This backbone network can then be used for various downstream tasks such as object detection, instance segmentation, and semantic segmentation (Chen et al., 2020). With this in mind, we examined SR for gaze estimation through the lens of self-supervision. In the context of GANs, the task of SR does not require any explicit labelling of the data since models are trained based on pairs of high and low resolution images. Due to this fact, we propose to obtain a backbone network trained for SR on a large unlabelled dataset and subsequently use it with a relatively simple head, trained on a small labelled gaze dataset. Intermediate representations extracted during the SR might prove useful for gaze estimation, resulting in a sample efficient gaze model that can achieve competitive results using less labelled training data.

### 4.1. SuperVision Architecture

We propose a novel end-to-end SR-Gaze architecture by leveraging self-supervised learning called "SuperVision". Concretely, using SwinIR-4x as the backbone and a simple ResNet18 as the head, the networks are concatenated to produce an end-to-end model for appearance-based gaze estimation as seen in Figure 3. The SwinIR-4x module is comprised of three parts, a shallow feature extractor, a deep feature extractor, and high-quality image reconstruction making for a total model size of approximately 27 million parameters (Wang et al., 2021a). The shallow feature extractor is comprised of a CNN and is used for retrieving global features such as overall colour and brightness while the deep feature extractor is comprised of a shifted window transformer architecture and is used for retrieving local features and small details. Lastly, high-quality image reconstruction uses the shallow and deep feature extractors to upscale and compress the data to a high-resolution image. SwinIR-4x uses the shallow feature extractor to pass information to the deep feature extractor, which in

turn passes the output to the high-quality reconstruction module. Additionally, the model uses a residual connection to concatenate shallow features with the deep features before the image reconstruction process. This image is directly passed to ResNet18 for gaze estimation, however, it is likely that auxiliary information is lost at the point in which SwinIR-4x compresses the features into the three colour channels associated with the output image. This bottleneck, coupled with SuperVision's depth, justifies using a residual connection from earlier layers in the SwinIR-4x module to later layers in ResNet18. This residual connection begins after the shallow and deep features are concatenated within SwinIR-4x. This residual connection skips the image reconstruction layers to connect at the second block of ResNet18. Finally, ResNet18 is used as the head for the task of gaze estimation and outputs the two-dimensional gaze vector.

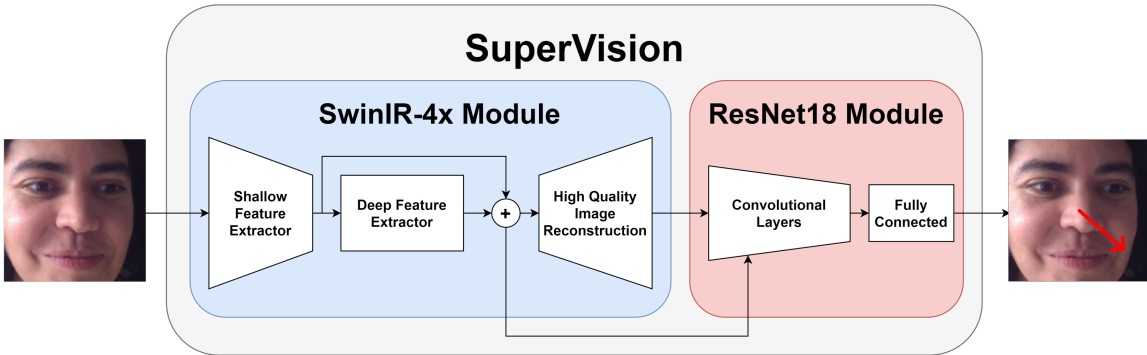

Figure 3: The SuperVision architecture takes an input image and outputs a gaze prediction. The model is comprised of two sub-modules where SwinIR-4x is used as the backbone network and ResNet18 as the head. There is a residual connection running from SwinIR's feature extractor to the second block in ResNet18.

## 4.2. Experimental Settings

To evaluate the efficacy of SuperVision on small datasets, we ran experiments to compare it with the aforementioned two-step SR framework and an interpolation baseline. All conditions used the same ResNet18 model for gaze estimation to avoid confounding variables. For this experiment, low-resolution images of size $112 \times 112$ were used so that the SwinIR-4x's upsampling outputs a reasonable $448 \times 448$ size. These images are then downsampled as ResNet18 accepts images of size $224 \times 224$. To accentuate differences even more, we opted to use the complexly degraded data provided by BSR-GAN's degradation model (Zhang et al., 2021). Additionally, these experiments are evaluated with 5%, 10%, and 20% of the labelled data. All hyperparameters were kept the same as in previous experiments except SuperVision's batch size, which had to be reduced to 4 due to model size and hardware constraints. This led to a training time of approximately two days on a Tesla v100 versus only a few hours for the simple ResNet18 models.

| Interpolation Type | Angular Error [°] | | |
|:---:|:---:|:---:|:---:|
| Bicubic Interpolation | 6.26 | 6.06 | 6.04 |
| SwinIR-4x Downsampled | 6.20 | 6.01 | 5.91 |
| SuperVision | **6.17** | **5.90** | **4.54** |
| Data Used | 5% | 10% | 20% |

Table 5: Training with a small percent of the training data using a simple ResNet18 as the gaze model. For reference, when using 100% training data, the angular error of GazeTR is 5.37° (see Table 4)

### 4.3. Results

As expected, the results in Table 5 show a trend of improvement as more labelled training data is used. It is particularly noteworthy that while the two-step-framework outperforms the baseline, both results have decelerating improvements in performance suggesting they are reaching a plateau. On the other hand, SuperVision's relative improvement over the interpolation baseline is accelerating as it achieved an angular error of 4.54° and demonstrated a 33% improvement over the baseline at 20% training data. Given the success with only 20% training data, the SuperVision model was also evaluated without a residual connection. The results indicated a 4.6% improvement with the inclusion of a residual connection, which supports the notion that there is a bottleneck between the SR and gaze modules.

Moreover, we can directly compare SuperVision's results to those of Table 4 since both models use the same data of $112 \times 112$ resolution with complex degradations. Despite using only 20% of the data, the SuperVision model achieved an angular error of 4.54° yielding a 15.5% improvement over the state-of-the-art model of GazeTR (angular error 5.37°). GazeTR had also been pretrained on the full ETH-XGaze dataset, and used 100% of the labeled data in MPIIFaceGaze dataset. These experiments demonstrate that the use of self-supervised learning on unlabelled data can be a good supplement when large datasets such as ETH-XGaze are not available. Furthermore, SuperVision is able to achieve high performance with only a fraction of labelled data from MPIIFaceGaze.

### 4.4. Visualizing SuperVision

The results from Table 5 are visualized in Figure 4 for two of the participants. The left column shows a low-resolution image of size $112 \times 112$, which is then degraded in the second column. The third column depicts a sample of images that were processed by SwinIR-4x as part of the two-step framework. The third column has been denoised considerably but there are some warped facial features, especially around the eyes. Lastly, by visualizing the output features of the SwinIR-4x module in SuperVision, we can see how the task of gaze estimation fine tunes the SwinIR module such that the output is no longer a high-resolution image. Furthermore, while the output of the SwinIR-4x module remains the same shape of a high-resolution image, it has been augmented such that it now represents nine unique grayscale images, each capturing different features of the input image. In contrast to the two-step framework, SuperVision is not as prone to the warping of facial features, which is likely attributed to the process of fine-tuning the SR and gaze modules end-to-end. One

possible explanation for this result could be that in the process of fine-tuning, SuperVision no longer needs to maintain a single colour image output and instead, optimizes for gaze estimation.

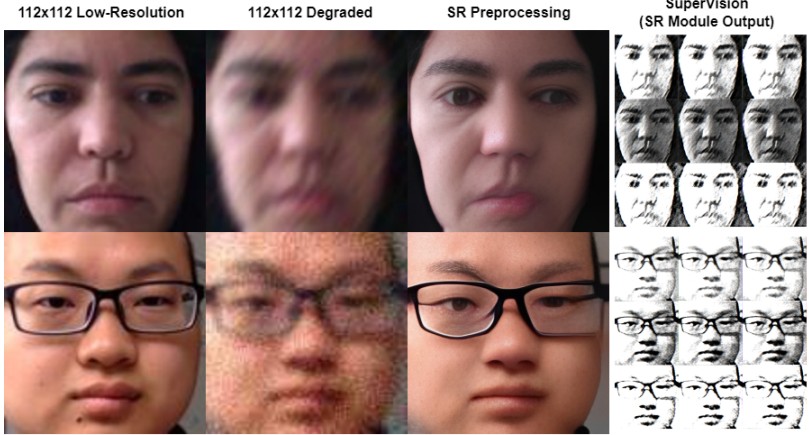

Figure 4: The left column is a low-resolution image, which is degraded in the second column. Using the degraded image as input, the third column visualizes the output of the SR step of the proposed two-step framework while the last column visualizes the output of the SR module in the SuperVision architecture.

## 5. Discussion

We explored the potential of SR as a preprocessing step for appearance-based gaze estimation. Our study demonstrated that not all SR models are effective in preserving gaze direction. However, the proposed two-step framework based on SR achieved state-of-the-art results on the MPIIFaceGaze dataset by exhibiting its capacity to enhance current appearance-based gaze estimation approaches. The method was also evaluated on low-resolution and degraded images and the results support the effectiveness of SR preprocessing. The utilization of SR with lower resolution images can be advantageous in fields like edge computing, as it allows one to bypass the requirement for high-resolution hardware. SR methods were also shown to be effective on degraded data (e.g., motion blur, video compression, and noise), which is often present in many computer vision datasets. Moreover, the newly introduced SuperVision architecture, which combines SR and ResNet18 with skip connections, showed impressive results while using only 20% of the labeled data, and in turn, outperformed the state-of-the-art GazeTR method by 15.5%. SuperVision demonstrated that self-supervised SR could reduce the need for large datasets for gaze estimation tasks. This is especially important for medical applications in which obtaining large datasets, such as from infants or older adults, is often difficult to collect. These findings demonstrate the potential of SuperVision as a more efficient and effective approach to appearance-based gaze estimation. To support the assertion that SR can serve as an efficient denoising tool with broad applicability, future studies could investigate the application of SR preprocessing to various other tasks.

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
