# OpenReview forum: "SuperVision: Self-Supervised Super-Resolution for Appearance-Based Gaze Estimation"
_NeurIPS.cc/2023/Workshop/Gaze_Meets_ML — Gaze Meets ML 2023 Oral_

### Official Review · Reviewer_a4nq · 2023-10-17
**Interesting insights about super-resolution for gaze estimation with good method contribution achieving SOTA performance**

**Rating:** 8
**Confidence:** 4

**Review:**

The authors first study the usefulness on of super-resolution (SR) for gaze estimation and demonstrate that some SR models help to improve the gaze estimation performance.
They then propose a new method for appearance based gaze estimation by training a light gaze estimation on top of a super resolution backbone.
They show that this novel method outperforms the current SOTA by quite a margin while requiring significantly less training data.

- Overall the paper is well written, nicely structured and clear.
- The experiments about evaluating the usefulness of SR models for gaze estimation and the subsequent analysis is very leads to interesting insights. I wish the authors would have compared more than only two super resolution models for this.
-  Would be interesting to have the performance numbers for more than just 20% of the data (Table 5

---

### Official Review · Reviewer_Pybt · 2023-10-24
**Interesting, well-explained, and thorough application of SR to gaze estimation**

**Rating:** 8
**Confidence:** 3

**Review:**

The methodology is well explained, clearly written, easy to read, with sufficient detail of experiments, datasets, and configurations.

Interesting exploration of SR for gaze estimation, with thorough evaluation and control for potential confounding factors.

The authors also mention computation time, system tested on, and comparison with prior works.

**Comments**

- Adding a figure of your approach would help the reader
- Is the code available?
- There is some extent of background information in the methodology section that shouldn't be there, e.g. beginning of section 4, focus on your methodology
- The authors should add limitations of the study and a more extensive description of potential future directions
- Table 5: Define “small portion” in the caption as well

---

### Meta-Review · Area_Chair_1mmZ · 2023-10-26

**Recommendation:** Accept (Oral)
**Confidence:** 5

**Metareview:**

The paper presents an interesting methodology to improve gaze estimation using super-resolution as preprocessing. The paper is well written and the methodology well explained and supported with comparative experiments achieving SOTA results. I highly recommend authors to try to address reviewers' comments to further improve the overall quality.

---

### Decision · Program_Chairs · 2023-10-26

Accept (Oral)